# Hypoxic Human Microglia Promote Angiogenesis Through Extracellular Vesicle Release

**DOI:** 10.3390/ijms252312508

**Published:** 2024-11-21

**Authors:** Alessandra Maria Testa, Livia Vignozzi, Diana Corallo, Sanja Aveic, Antonella Viola, Manuela Allegra, Roberta Angioni

**Affiliations:** 1Department of Biomedical Sciences, University of Padua, 35131 Padua, Italy; alessandramaria.testa@unipd.it (A.M.T.); livia.vignozzi@unipd.it (L.V.); antonella.viola@unipd.it (A.V.); 2Laboratory of Immunity, Inflammation and Angiogenesis, Istituto di Ricerca Pediatrica (IRP), Fondazione Città della Speranza, 35127 Padua, Italy; 3Laboratory of Target Discovery and Biology of Neuroblastoma, Istituto di Ricerca Pediatrica (IRP), Fondazione Città della Speranza, 35127 Padua, Italy; d.corallo@irpcds.org (D.C.); s.aveic@irpcds.org (S.A.); 4Laboratory of Neuronal Circuits in Developmental Disorders, Istituto di Ricerca Pediatrica (IRP), Fondazione Città della Speranza, 35127 Padua, Italy; manuela.allegra@unipd.it; 5Neuroscience Institute, National Research Council, 35131 Padua, Italy

**Keywords:** extracellular vesicles, angiogenesis, microglia, hypoxia, neuroinflammation, stroke

## Abstract

Microglia, the brain-resident immune cells, orchestrate neuroinflammatory responses and are crucial in the progression of neurological diseases, including ischemic stroke (IS), which accounts for approximately 85% of all strokes worldwide. Initially deemed detrimental, microglial activation has been shown to perform protective functions in the ischemic brain. Besides their effects on neurons, microglia play a role in promoting post-ischemic angiogenesis, a pivotal step for restoring oxygen and nutrient supply. However, the molecular mechanisms underlying microglia–endothelial cell interactions remain largely unresolved, particularly in humans. Using both in vitro and in vivo models, we investigated the angiogenic signature and properties of extracellular vesicles (EVs) released by human microglia upon hypoxia–reperfusion stimulation. EVs were isolated and characterized in terms of their size, concentration, and protein content. Their angiogenic potential was evaluated using endothelial cell assays and a zebrafish xenograft model. The in vivo effects were further assessed in a mouse model of ischemic stroke. Our findings identified key proteins orchestrating the pro-angiogenic functions of human microglial EVs under hypoxic conditions. In vitro assays demonstrated that hypoxic EVs (hypEVs) promoted endothelial cell migration and tube formation. In vivo, hypEVs induced vessel sprouting in zebrafish and increased microvessel density in the perilesional area of mice following ischemic stroke.

## 1. Introduction

Microglia, the brain-resident immune cells, are key orchestrators of the neuroinflammatory response with implications in several neurological diseases, including brain ischemia [1]. Ischemic stroke (IS) accounts for about 85% of all strokes globally, representing a major cause of mortality and chronic morbidity [2]. While initially regarded as detrimental to stroke progression, numerous studies have nowadays demonstrated that microglial activation performs essential protective functions in the ischemic brain [1,3]. Indeed, microglia were shown to limit astrocyte activation, neutrophil infiltration, and neuronal excitotoxicity, thus favouring neuroplasticity and preserving myelin integrity [4,5,6,7]. In addition to the neuronal effect, substantial evidence points to an active role of microglia in regulating post-ischemic angiogenesis [1,8]. Importantly, the vascular remodelling in the perilesional area upon an ischemic event is crucial for restoring oxygen and nutrient supply, thus eventually facilitating neuroplasticity [9]. In line with this, modulating microglial activities in the ischemic brain has been reported to enhance positive outcomes, through the local stimulation of angiogenesis [10,11]. Accordingly, depletion or inhibition of microglia leads to impaired angiogenesis in preclinical stroke models [12,13]. However, the specific molecular components mediating the microglial beneficial effect on the ischemic vasculature are still undefined.

Recently, it has been reported that microglia may engage in intricate interactions with other cells by releasing extracellular vesicles (EVs) [14]. EVs represent important cell-to-cell communication vehicles in the central nervous system (CNS), modulating brain homeostasis and plasticity [15,16]. Here, microglial EVs modulate neuronal functionality and synaptic transmission through several molecular mechanisms [14]. Upon exposure to pro-inflammatory stimuli, such as bacterial lipopolysaccharide (LPS) or ATP, microglia release EVs enriched in IL1-β, which sustain the neuroinflammatory response to tissue damage [17,18,19]. ATP also stimulates microglia to release EVs enriched in regulators of cell metabolism, cell adhesion, and extracellular matrix remodelling, in addition to proteins involved in pro-inflammatory signalling [20]. These reports highlighted the importance of the EV-based communication between microglia and other cellular compartments within the injured brain.

Overall, this growing evidence demonstrates (a) the implication of microglia in ischemic progression [5], (b) the importance of the microglia–vascular network for improving disease outcomes [8,11], and (c) the microglial release of EVs for intercellular communication [14]. Strengthened by this evidence, we hypothesised that under ischemic conditions, human microglia release pro-angiogenic EVs to promote vascular remodelling. To test this hypothesis, we analysed the signature and the functional properties of EVs released by human microglia upon hypoxia–reperfusion stimulation, using both in vitro and in vivo models of angiogenesis. Our study identifies a specific pro-angiogenic signature of EVs released by human microglial cells under hypoxic conditions. 

## 2. Results

### 2.1. In Vitro OGD/R Stimulation of the Human Microglial Cell Line HMC-3 Mimics Stroke-Induced Activation

To evaluate microglial activation in response to the hypoxic insult proper of IS, we set up an in vitro protocol using the human microglial cell line HMC-3. To mimic the stroke microenvironment, cells were stimulated with an oxygen–glucose deprivation/reperfusion (OGD/R) protocol [21], as represented in Figure 1A. Cell viability was assessed by flow cytometry, to exclude inappropriate toxicity of the OGD/R protocol (Figure 1B). Annexin V staining revealed that hypoxic stimulation did not significantly affect HMC-3 viability, either after 3 h (h) of OGD or following 24 h reperfusion (OGD/R), compared to cells cultured under normoxic conditions (Figure 1B). The ability of the OGD/R protocol to modify the microglial transcriptional signature in response to the hypoxic insult, similarly to the in vivo scenario, was confirmed by the increased expression of known hypoxia-induced genes, namely vascular endothelial growth factor (*VEGFA*), glucose transporter 1 (*GLUT1*) and growth differentiation factor 15 (*GDF15*), in OGD- and OGD/R-treated cells, compared to their expression levels under control conditions (Figure 1C–E). Moreover, the assessment of microglia activation by real-time polymerase chain reaction (qRT-PCR) revealed that the pro-inflammatory marker *IL1B*, as well as the anti-inflammatory genes *ARG1*, *MRC1*, and *IL10*, were upregulated after OGD compared to control levels (Figure 1F–L). To note, *IL1B* expression levels were restored after the reperfusion. Instead, although no differences were observed for *TNFAlpha* (*TNFA*) and *TGFB1* after OGD stimulation alone, their expression was significantly induced after reperfusion, compared to unstimulated cells (Figure 1F–L). Taken together, these data confirmed that the OGD/R stimulation shapes the human microglial cell phenotype, resembling in vivo activation responses.

### 2.2. EVs from Hypoxic Microglia Promote Angiogenesis In Vitro and in a Zebrafish Xenograft Model

The morphological and biological properties of microglia-secreted EVs were defined by collecting the conditioned medium (CM) from HMC-3 cells exposed to hypoxia (hyp-HMC-3-CM), at the end of the previously described protocol (OGD/R), or from unstimulated HMC-3 cells as a control (contr-HMC-3-CM). Hypoxic EVs (hypEVs) and control EVs (contrEVs) were then isolated by ultrafiltration (REF) from the hyp-HMC-3-CM and contr-HMC-3-CM, respectively. Subsequent validation of the EV isolation procedure was performed accordingly with MISEV 2023 guidelines [22], firstly by estimating the size of purified particles by TEM and Nanosight (Figure 2A,C,D). Here, contrEVs and hypEVs displayed a similar size (mean sizes of 141.5 ± 3.4 nm and 134.0 ± 5.4 nm, respectively) and concentration (8.78 × 10^8^ ± 7.83 × 10^7^ particles/mL and 4.10 × 10^8^ ± 7.72 × 10^7^ particles/mL, respectively). Then, the expression of conventional EV markers (CD9 and CD63) was validated by Western blot (Figure 2B), confirming the quality of the EVs derived from HMC-3 cells following our stimulation protocol. Therefore, we tested the angiogenic properties of EVs isolated from hypoxia-activated human microglia. To this aim, a series of in vitro assays were exploited, using a human brain microvascular cell line, the HBEC-5i. We first assessed whether microglial EVs could directly target brain ECs through an uptake assay (Figure 3A,B). Here, EVs were stained with the Dil lipophilic dye and then administered to HBEC-5i cells in culture. After 6 h of incubation with the Dil^+^ EVs, the percentage of fluorescent ECs was analysed by flow cytometry. Our data indicated that both contrEVs and hypEVs were taken up by HBEC-5i cells (Figure 3A,B). Once the direct interaction between brain ECs and microglial EVs was highlighted, the ability of microglial EVs to affect EC migration was evaluated in vitro through a scratch wound healing assay (Figure 3C,D). As expected, the migration of HBEC-5i cells was increased upon the stimulation with human recombinant VEGF (rhVEGF). Similarly, hypEVs enhanced endothelial migration compared to untreated cells. In contrast, contrEVs did not significantly affect endothelial migration. Then, we checked whether microglial EVs could further influence the ability of ECs to form capillary-like structures in vitro, using a tube formation assay (Figure 3E,F). Treatment with hypEVs significantly enhanced capillary network formation, as shown by the quantification of the total length of the tubules. Again, contrEV treatment did not significantly affect the process. Moreover, hypEV treatment enhanced the endothelial ability to digest the extracellular matrix (ECM), while no effect of the EVs was observed on endothelial cell proliferation (Appendix A–C).

The pro-angiogenic activity of hypEVs was further validated in vivo in a zebrafish xenograft angiogenesis model (Figure 4). Briefly, contrEVs, hypEVs, and the vehicle (PBS) were resuspended in 75% Matrigel and injected in the perivitelline space of the yolk sac of 48 h post-fertilisation (hpf) zebrafish embryos. This strategy allowed the graft to be placed in the proximity of the developing sub-intestinal vein (SIV), as depicted in Figure 4A, thus prompting its de novo vascularization. The effect of each treatment on vascular development dynamics was assessed at 1 day post-injection (1 dpi) by alkaline phosphatase (AP) staining (Figure 4A). As shown in Figure 4B,C, hypEVs significantly increased the number of branches sprouting from the SIV towards the implant, compared to contrEVs. Notably, the injection of Matrigel alone did not affect vascular sprouting. According to our in vitro data, this result demonstrated a rapid pro-angiogenic effect of hypEVs in the zebrafish in vivo model.

### 2.3. Combined Activities of ENG, HGF, and ARTN Mediate the hypEVs Pro-Angiogenic Effect In Vitro

To identify the molecular mediator/s of the hypEVs pro-angiogenic action, the protein cargo of contrEVs and hypEVs was analysed, using the Protein Profiler Human Angiogenesis Array Kit (Figure 5). Data revealed that, among all analytes, uniquely artemin (ARTN), endocrine gland-derived vascular endothelial growth factor, also known as prokineticin 1 (EG-VEGF/PROK1), endoglin (CD105/ENG), endostatin/collagen type XVII alpha 1 (END/COL18A1), hepatocyte growth factor (HGF), and interleukin-8 (IL-8) were significantly enriched in hypEVs compared to contrEV preparations (Figure 5A–C). Additional factors, namely insulin growth factor binding protein 1 (IGFBP1), matrix metalloproteinase 9 (MMP-9), leptin (LEP), and placental growth factor (PIGF) were significantly down-regulated in hypEVs compared to contrEVs (Figure 5A,B). Focusing on the highest expressed pro-angiogenic analytes, we further validated their up-regulation at the transcriptional level. Thus, we confirmed by qRT-PCR that the *ARTN, HGF, ENG,* and *PROK1* genes were significantly induced in HMC-3 cells exposed to OGD/R, compared to unstimulated cells, with different time-dependent dynamics (Appendix A–D). Importantly, overexpression of the *ARTN, HGF, ENG* genes were equally detected in primary mouse microglial cells exposed in vitro to the OGD/R protocol (Figure 5C–F and Appendix A). A trend for increased *PROK1* expression was also present upon ex vivo OGD/R exposure, although not significant (Figure 5F).

Once the angiogenic cargo of hypEVs was defined, we investigated whether the enrichment in specific proteins in EVs was functionally associated with the angiogenesis induction in brain ECs. Thus, we employed commercially available neutralising antibodies against the factors that were significantly up-regulated in both the microglial cell line and in the ex vivo stimulated cells (ARTN, ENG, and HGF), to investigate whether blocking the activity of any of these molecules, alone or in combination, might alter the hypEV angiogenic properties. In the tube formation assay, the individual or two-by-two blocking of ARTN, ENG, or HGF carried by hypEVs did not significantly impair their pro-tubulogenesis effect (Figure 6A–D). However, the mixture of all antibodies led to a significant inhibition of hypEV-induced angiogenesis, suggesting a combined contribution of the ARTN, ENG, and HGF proteins for the angiogenic action of hypEVs in vitro (Figure 6E).

### 2.4. EVs from Hypoxic Microglia Promote Vessel Remodelling in a Mouse Model for Ischemic Stroke

To evaluate whether hypEVs could modulate the angiogenic process triggered by an ischemic injury, we employed a mouse model for ischemic stroke (Figure 7). To recapitulate neurovascular and neuroplastic rearrangements typical of the ischemic penumbra, we used the middle cerebral artery occlusion (MCAO) technique on 3-month-old mice (*n* = 5 PBS, *n* = 6 hypEVs). Intraperitoneal injections of hypEVs or PBS were then performed immediately after the surgery, and each 24 h, for 5 consecutive days. To evaluate the microvessel density in the perilesional area (M1), mice were perfused at day 7 post-MCAO and immunofluorescence for the endothelial marker CD31 was performed on brain sections. We observed a significant increase in the vessel density in the perilesional areas in mice treated with hypEVs, compared to PBS-injected mice, as shown by the quantification of the percentage of the CD31^+^ area for each field of view (Figure 7A,B). To complement our quantification of microvessel remodelling, we evaluated cell proliferation activity within the perilesional area, by immunostaining for the proliferation marker Ki67. This analysis was performed at day 7 post-MCAO on brain sections from both PBS and hypEV-treated animals (Figure 7C). Our findings indicated that animals receiving hypEVs show a modest, yet significantly increased number of proliferating (Ki67^+^) cells compared to controls, with counts per field of view ranging from 0 to 4. The percentage of the Ki67^+^ area was also increased in hypEV-treated animals. Additionally, while CD31^+^Ki67^+^ cells were hardly visualised in the area of interest, their occurrence was higher in animals treated with hypEVs compared to controls, suggesting a potential contribution to the angiogenic process. This approach does not account for cumulative proliferative activity over the 7 days post MCAO, capturing only actively proliferating cells at a single time point. However, our findings indicated that hypEVs may enhance both general cell proliferation and endothelial presence in the ischemic area at day 7 post-stroke. In line with our previous analyses, these data provide evidence that hypEVs exert a pro-angiogenic action on brain ECs following an ischemic injury in vivo.

## 3. Discussion

Ischemic stroke (IS) accounts for about 85% of all strokes globally, representing a major cause of mortality and chronic morbidity. It represents a critical medical emergency marked by the rapid focal neurological deficits within a specific vascular region. Despite advances in medical science, effective therapies remain limited, making it crucial to dissect the pathophysiological mechanisms underlying this condition to develop better treatment strategies and improve patient outcomes. In this scenario, growing evidence highlights the critical role of proper vascular remodelling in the functional recovery of the post-ischemic brain. Microglia are central to this process, shaping the microenvironment and supporting angiogenesis. However, the precise mechanisms behind these events in cerebral ischemia remain largely elusive.

This study reveals an EV-based communication between human microglial and endothelial cells within the brain. We demonstrated that, under hypoxia–reperfusion, HMC-3 human microglial cells undergo a hypoxia-specific activation in vitro that results in a positive regulation of pro- and anti- inflammatory genes. Consistent with our data, recent murine brain analyses at the single-cell level noted multiple stroke-responsive microglial subclusters, overexpressing specific sets of pro-inflammatory or reparative molecules [23]. Not surprisingly, this microglial hypoxia-induced phenotype is associated with a distinct release of EVs. Hypoxia is already known to affect the composition and biological functions of EVs across various cell types [24,25]. Indeed, extensive literature has shown the pivotal role of oxygen deprivation in making up different EV faces from several cells, resulting in altered autophagy, survival, proliferation, metabolism, and motility of EV-target cells [26]. In the cerebral environment, a recent study demonstrated that brain-derived EVs (BDEVs) secreted in the 72 h following experimental stroke are enriched in a wide array of mRNAs encoding proteins involved in inflammatory and reparative processes [27]. In line with this, EVs from murine microglial cells exposed to in vitro OGD were reported to install a feedback circuit, enhancing microglial M2 polarisation and attenuating astrogliosis in mice with experimental stroke [28]. Moreover, microglia isolated from the injured hemisphere of neonatal stroke models are reported to secrete EVs that are more efficiently internalised by other microglial cells, thus suggesting an influence on microglial EV uptake dynamics [29]. However, the potential impact of hypoxia on the established interaction between microglia and the endothelium remains to be elucidated. Specifically, it is unclear whether hypoxia influences the release of vesicles from human microglia with distinct angiogenic properties.

Here, we also confirmed that human microglial EVs directly target the endothelium, mimicking the murine context. Indeed, to our knowledge, this is the first study assessing the angiogenic properties of microglial EVs using an experimental system where all cellular components, both microglia and endothelial cells, are of human brain origin. Furthermore, we demonstrated that this EV angiogenic ability is shaped by the hypoxic microenvironment. Indeed, unlike EVs released in normoxic conditions, EVs released by hypoxic microglial cells (hypEVs) promote endothelial cell migration and tube formation in vitro. Accordingly, hypEVs induce vessel sprouting in a zebrafish xenograft angiogenesis model, demonstrating specific hypoxia-induced pro-angiogenic activity. The characterization of the angiogenic cargo of hypEVs revealed an enrichment in specific factors, namely endoglin (ENG), hepatocyte growth factor (HGF), artemin (ARTN), and prokineticin-1 (PROK1), known inducers of angiogenesis in various tissues and conditions. ENG, a transmembrane glycoprotein highly expressed in activated ECs, monocytes, fibroblasts, and mesenchymal and smooth muscle cells, acts as a co-receptor for the TGF-β family of ligands, by modulating signalling through two type I TGF-β receptors (TβRI), namely activin-like kinase-1 (ALK-1) and ALK-5, or by activating TGF-β-independent pathways [30,31]. To note, ENG plays a crucial role in vascular development and maintenance, and it is overexpressed during hypoxia, wound healing, and inflammation [30,32]. Various studies revealed the role of ENG in preventing hypoxia-induced EC death and in promoting vascular stability, while ENG deficiency was linked with poorer outcomes in stroke models, demonstrating its major role in post-ischemic vascular remodelling and recovery [30,31]. HGF, a pleiotropic cytokine promoting cell proliferation, survival, motility, and morphogenesis, acts by binding to the receptor tyrosine kinase (RTK) type receptor cMET, expressed in most body districts, including the liver, pancreas, bone marrow, muscle, kidney, and brain [33,34]. Notably, the expression of HGF and its receptor are induced upon hypoxia and tissue injury, where HGF signalling fosters tissue repair through anti-apoptotic, anti-inflammatory, anti-fibrotic and pro-angiogenic mechanisms [34]. Moreover, due to its major neuroprotective actions, HGF-based preclinical therapeutic strategies have been investigated in experimental stroke models [35]. Interestingly, an increased EV loading of ENG and HGF was consistently observed in hypoxia-preconditioned mesenchymal stromal cells (MSCs) and associated to their pro-angiogenic potential [36]. As far as ARTN is concerned, although its overexpression under hypoxia has been reported, its roles in stroke are unclear [37]. ARTN, a member of the glial cell-derived neurotrophic factor (GDNF) family of ligands, promotes neuronal differentiation and survival, primarily through binding to the GDNF receptor alpha (GFRα) family, with highest affinity for GFRα3, leading to activation of the RET receptor tyrosine kinase (RTK) [38]. In addition, recent studies have suggested a role for ARTN in tumour invasiveness and angiogenesis [39]. PROK1 is a pleiotropic factor secreted by endocrine glands. Since we could not confirm a significant PROK1 overexpression in primary microglia, we did not extend our further investigations to this factor. Our results demonstrate that microglial EVs released under hypoxic conditions are enriched in ENG, HGF, and ARTN, and support their combined role as mediators of the EV pro-angiogenic effects towards brain ECs in our system.

In highlighting the intricate molecular network underlying microglia–vascular interactions in a hypoxic environment, we must mention that, recently, a work by Zhang et al. has shown that EV-associated TGF-β plays a non-redundant role in brain angiogenesis induced by mouse BV-2 microglial cells [40]. Although we confirmed *Tgfb1* overexpression in primary murine microglia exposed to OGD/R in vitro (Appendix A), TGF-β1 was not enriched in human hypEVs (Figure 5A and Appendix A) and the use of a neutralising antibody to block EV-associated TGF-β1 did not impair tube formation in vitro (Appendix A). Thus, although we do not exclude a role for TGF-β1 in human hypEV-induced angiogenesis, our data suggest that the pro-angiogenic action of human EVs released by hypoxic microglial cells depends on multiple pathways and involves ARTN, ENG, and HGF, which are specifically and consistently up-regulated in human microglial EVs showing pro-angiogenic properties. Unfortunately, the current technological limitations in visualising and analysing microglial EVs in situ hinder the complete validation of our hypothesis in vivo. Despite this challenge, the observation that primary microglia, isolated from the murine brain and exposed to OGD/R, up-regulate the gene expression of the identified molecules (*ART, ENG, and HGF*) suggests that these mediators may also be involved in microglial responses to ischemic injuries in vivo. Next, we aim to conduct spatial transcriptomics ex vivo on the ischemic mouse brain, which will allow us to investigate the pathophysiological involvement of the endothelium under brain-ischemia conditions. Further experimental approaches will help us understand endothelial responses providing insights into potential beneficial post-injury effects.

To corroborate the relevance of the pro-angiogenic effects of human microglial EVs in a disease-specific context, a murine stroke model was exploited. The occlusion of the middle cerebral artery (MCAO) in adult mice results in an ischemic injury closely resembling that seen in human stroke patients [41]. This well-established model replicates the complex biological responses observed in the human brain, including neurovascular and neuroplastic rearrangements, making it a valuable system for evaluating treatment effects on perilesional repair processes, including angiogenesis [41]. Following MCAO surgery, hypEVs or PBS were administered intraperitoneally (IP) to minimise post-operative stress and complications associated with intravenous EV injection. While we acknowledge that we have not compared the biodistribution of IP-injected EVs in other organs (e.g., liver, lungs, and intestines), previous studies, including ours, suggest that IP administration can effectively deliver exosomes to target tissues, particularly in contexts of active angiogenesis [42]. Although we cannot claim IP is superior to IV in terms of efficiency of brain accumulation, in our experimental setup, IP injection offers a practical and effective compromise between feasibility and efficacy [42,43,44]. The administration of hypEVs increased the vessel density in the perilesional area, further validating their effect on vascular remodelling after an ischemic injury. Previous studies employing the MCAO model suggested that modulation of endogenous microglia could exert an important local action by promoting vascular remodelling, eventually ameliorating stroke outcomes [11,45]. Moreover, transplantation of microglia polarised in vitro towards anti-inflammatory phenotypes was found to have beneficial effects on vessel repair and disease progression in murine stroke models [46,47]. These studies were instrumental in establishing the significance of promoting vascular remodelling and tuning microglial functions to aid brain recovery after stroke. However, they did not elucidate the role of human microglia exposed to an ischemic environment on vasculature function, emphasising the importance of our investigation. This experiment, besides validating the potential for using microglial EVs as therapeutic tools, reinforces the crucial role of microglia in stroke pathophysiology and of EVs as a critical communication system. Taken together, the significance of our findings extends to a deeper understanding of the molecular bases of microglial interactions with the vasculature in a human context, essential for grasping their unique roles in stroke progression.

Overall, this study identifies EVs released by human microglia upon hypoxia–reperfusion as active promoters of angiogenesis, providing novel mechanistic insights into how microglia drive this essential process in stroke pathophysiology. Moreover, it opens avenues for additional research on potential strategies to target microglia and their EVs for enhanced stroke recovery. The exploitation of EVs in clinical applications holds great promise, particularly in the development of EV-based therapies to stimulate angiogenesis. This approach could lead to innovative treatments that harness the natural capabilities of EVs for tissue repair and regeneration. Dissecting these mechanisms of actions is crucial to deepen our understanding of ischemic pathophysiology and of the contributions of microglia to vascular repair. These insights could benefit the study of biomarkers for improved diagnosis, prognosis, and follow-up in ischemic diseases, leading to more personalised and timely interventions. Moreover, delineating these pathways could help develop more effective treatments for conditions such as stroke, where promoting vascular remodelling and regeneration is critical for recovery. Ultimately, uncovering the signalling mechanisms underlying microglia-driven angiogenesis holds significant promise for advancing both fundamental research and clinical practice in vascular medicine.

## 4. Materials and Methods

### 4.1. Cell Lines and Stimulation Protocols

HMC-3 human microglial cells (ATCC, Manassas, VA, USA, #CRL-3304) were cultured in a humidified incubator with 5% CO_2_ at 37 °C, in EMEM (ATCC, Manassas, VA, USA, #302003) supplemented with 10% heat-inactivated foetal bovine serum (FBS) (ATCC, #302020), and 1% penicillin and streptomycin. HMC-3 were subcultured when approximately 80% confluent by trypsinization. HBEC-5i human brain microvascular cells (ATCC, Manassas, VA, USA, #CRL-32-45) were cultured in 0.1% gelatin-coated flasks or plates in a humidified incubator with 5% CO_2_ at 37 °C, in DMED:F12 (ATCC, Manassas, VA, USA, #30-2006) supplemented with 10% FBS (GE Hyclone, Fisher Scientific, Segrate, Italy, #SV30160.03), and 1% penicillin and streptomycin, with the addition of 30 mg/L Endothelial Cell Growth Supplement (ECGS, Corning, Bedford, MA, USA, #354006). For EV isolation, cells were cultured in T75 flasks and OGD/R stimulation was performed at 80% confluence. For OGD, cells were washed once with phosphate-buffered saline (PBS) 1×, incubated in glucose-free medium (DMEM, no glucose, Gibco, Paisley, UK, #11966025), supplemented with 1% penicillin and streptomycin, in the absence of FBS, and transferred to a hypoxic incubator chamber, with 1% O_2_, 5% CO_2_ at 37 °C. For reperfusion (R) and subsequent conditioned medium (CM) collection for EV isolation, cells were cultured in a 5% CO_2_, 37 °C incubator in standard cell culture medium, in the absence of FBS, for 24 h. Normoxia controls were kept at 37 °C, 5% CO_2_. The collected CM was cleared of cell debris by centrifugation at 1000× *g* for 15 min (min), followed by CM preservation at −80 °C, prior to EV isolation.

### 4.2. EV Isolation and Characterization

EVs were isolated from control or hypoxia-preconditioned HMC-3 cell supernatants by ultrafiltration, using Amicon Ultra 15 mL Filters with a 100 kDa molecular weight cutoff (Merck Millipore, Darmstadt, Germany, #UFC910096), according to the manufacturer’s instructions. Briefly, each filter was first sterilised with 10 mL of 70% EtOH, followed by centrifugation at 4000× *g* for 5 min. Filters were washed twice with sterile PBS 1× (4000× *g*, 5 min). Subsequently, 12 mL of debris-cleared CM (derived from 3 × 10^6^ HMC-3 cells), were loaded onto each filter and centrifuged at 2800× *g* for 20 min, at room temperature (RT). The flow-through was discarded and 10 mL sterile PBS 1× were added to the filter. After spinning at 2800× *g* for 20 min, the concentrated EVs in about 150 µL of PBS were collected as single aliquots, followed by storage at −80 °C. EVs were thawed on ice and kept at 4 °C prior to use in downstream experiments, avoiding repeated freeze–thaw cycles. For Western blot analysis, EVs were lysed with 0.4% SDS in PBS, for membrane denaturation. The total protein content was quantified by Micro BCA Protein Assay Kit (Pierce, Thermo Fisher Scientific, Waltham, MA, USA#23235), according to the manufacturer’s instructions. Thus, 3–5 μg input protein were denatured with 4× Bolt LDS Sample Buffer (Thermo Fisher Scientific, Carlsbad, CA, USA, #B0007), followed by incubation at 95 °C for 5 min. Samples were separated by electrophoresis using a Bolt 4–12% Bis-Tris Plus Gel (Invitrogen, Carlsbad, CA, USA, #NW04125BOX), at 120 V for 1 h, using MOPS Running Buffer (Thermo Fisher Scientific, Carlsbad, CA, USA, #B0001), under reducing conditions, adding dithiothreitol (DTT) 10 mM (for LAP/TGF-β1 detection) or under non-reducing conditions (for CD63 and CD9 detection). Proteins were transferred onto polyvinylidene fluoride (PVDF) membranes using Bolt Transfer Buffer 10× (Thermo Fisher Scientific, Carlsbad, CA, USA, #BT00061), at 20 V for 1 h, at RT. Membranes were blocked in TBS 1×, 0.2% Tween, and 3% BSA and incubated o.n. at 4 °C with anti-CD63 (Abcam, Cambridge, UK, #59479), anti-CD9 (Biolegend, San Diego, CA, USA, #312102), or anti-LAP/TGF-β1 (R&D Systems, Minneapolis, MN, USA, #AF-246-NA) primary antibodies, diluted 1:1000 in TBS 1×, 0.2% Tween, and 1% BSA. Membranes were further incubated with the appropriate peroxidase-conjugated secondary antibody, anti-mouse (BioRad, Hercules, CA, USA, #170-6516), anti-rabbit (Cytiva, Fisher Scientific, Segrate, Italy, #NA934V), or anti-goat (Biorad, Hercules, CA, USA, #172-1034), for 1 h at RT. Chemiluminescence was achieved by ECL Prime Western Blot Detection Reagent (Cytiva, Milan, Italy, #RPN2232). Images were acquired using the iBright Imaging System (Invitrogen, Carlsbad, CA, USA). For transmission electron microscopy (TEM) visualisation, 20 µL of EVs in PBS were dispensed for 2 min on 300 mesh carbon-coated copper grids that were made hydrophilic by a 15 exposure to a glow discharge. A filter paper (Whatman, Maidstone, UK) was used to remove the excess of liquid. EVs were then stained with 1% uranyl acetate for 2 min. An FEI Tecnai G2 transmission electron microscope operating at 100 kV, with a Veleta (Olympus Soft Imaging System, Münster, Germany) digital camera, was exploited to capture images. With nanoparticle tracking analysis (NTA), particles were tracked and sized based on their Brownian motion and their diffusion coefficient. A total of 300 µL of undiluted EVs were used for each analysis. Samples were analysed under constant flow conditions (flow rate = 50) at 25 °C. At least 3 × 60 s successive videos were captured with a camera level of 11, generating at least 3 replicate histograms. Data were analysed using the NTA 3.1.54 software, which provides a merged analysis of all videos acquired for each sample, giving the mean, mode, and estimated concentration for each particle size. A detection threshold of 4 was employed for video analysis for all samples. In functional experiments where EVs were administered to HBEC-5i cells, EVs were used at a final concentration of 20 µg/mL.

### 4.3. Apoptosis Assay

A total of 1 × 10^5^ HMC-3 cells were seeded in each well of a 24-well plate and cultured o.n. After OGD or OGD/R treatment, cells and their supernatants were collected in the same tube. Normoxic cells treated with 2 μM Staurosporine (Sigma-Aldrich, Merck Life Science, Milan, Italy, #S4400) for 24 h were used as a positive apoptosis control. Samples were stained with Annexin V–APC antibody (eBioscience, Thermo Fisher Scientific, Wien, Austria, #BMS306APC-100), diluted 1:100 in 50 µL of Annexin V Binding Buffer (BD, Fisher Scientific, Segrate, Italy, #556454), for 30 min at 4 °C. Cells incubated in 50 µL buffer in the absence of Annexin V–APC antibodies were used as a negative control for flow cytometry. A wash in the binding buffer was performed, followed by resuspension in PBS 1×. Labelled cells were detected by fluorescence-activated cell sorting (FACS) at FACS Celesta. Data analysis was performed with the FlowJo software (v.10.9.0, 5 May 2023).

### 4.4. RNA Isolation, cDNA Synthesis, and RT-qPCR

For RNA extraction, 1 × 10^5^ HMC-3 cells were plated in 12-well plates and cultured o.n., prior to OGD/R experiments. After stimulation, cells were lysed with 300 µL Trizol per well. Nucleic acids were solubilized by adding 60 µL chloroform, followed by repeated inversions of the samples. Samples were centrifuged at 12,000× *g* at 4 °C for 30 min, and the RNA-containing aqueous phase was transferred to a new tube. A total of 400 µL isopropanol was added and samples were centrifuged for 30 min at 4 °C, 12,000× *g*. Supernatants were completely removed, and the pellet was washed in 500 µL 70% EtOH in diethyl pyrocarbonate (DEPC)-treated water, followed by centrifugation for 5 min at 4 °C, 12,000× *g*. Pellets were allowed to dry before rehydration with 21 μL of water. RNA concentration and quality were measured with a NanoDrop spectrophotometer. Retrotranscription was performed using the High-Capacity cDNA Reverse Transcription Kit (Thermo Fisher Scientific, Carlsbad, CA, USA, #4368813), according to the manufacturer’s instructions, using 500 ng of input RNA. Retrotranscription was performed with a T100tm Thermal cycler according to the following cycle: 10 min at 25 °C, 2 h at 37 °C, 5 min at 85 °C, and 4 °C infinite hold. Obtained cDNA samples were diluted 1:5 in DEPC-treated water. qPCR reactions were performed with Power Sybr Green PCR Master mix (Thermo Fisher Scientific, Carlsbad, CA, USA, #43-687-02), in a 384-well plate and run in a QuantStudio 5.0 thermocycler (Applied Biosystem-Thermo Fisher Scientific, Carlsbad, CA, USA). The reaction mix was prepared according to the manufacturer’s instructions, using 4 μL of cDNA sample, 4 μL of Power Sybr Green PCR Master Mix, 0.075 μL forward primers (100 µM), and 0.075 μL reverse primers (100 µM). MilliQ water was used as a negative control. Each reaction was run in technical duplicates. All primers used for qRT-PCR are listed in Table 1.

### 4.5. Uptake Assay

EVs were fluorescently labelled using Dil Stain (Invitrogen, Carlsbad, CA, USA, #D282) lipophilic membrane dye, by incubating purified EVs resuspended in 0.5 mL PBS 1× with 1 µM Dil, for 1 h at 4 °C, in the dark. Clean PBS without EVs treated with the same procedure alongside EV samples was used as a negative staining control. Labelled EV (EV^Dil^) and PBS (PBS^Dil^) samples were loaded onto single Exosome Spin Columns (MW3000) (Thermo Fisher Scientific, Carlsbad, CA, USA, #4484449), to remove excess dye, following the manufacturer’s instructions. EVs in the obtained final volume were quantified using the Qubit™ Protein and Protein Broad Range (BR) Assay Kit (Thermo Fisher Scientific, Carlsbad, CA, USA, #Q33212), prior to administration to cells. For the EV uptake assays, 0.8 × 10^5^ HBEC-5i cells were plated in each well of a 24-well plate, pre-coated with 0.1% gelatin in PBS 1×. EVs^Dil^ were added and after 6 h, cells were washed in PBS, detached with StemPro Accutase (Thermo Fisher Scientific, Carlsbad, CA, USA, #A11105-01), spun down for 10 min at 300× *g*, at RT, and resuspended in 200 µL PBS. Control cells were treated with PBS^DIL^. Cell fluorescence was detected at FACS Celesta and data were analysed using the FlowJo Software, to quantify the percentage of PE^+^ cells over the total cell number.

### 4.6. Scratch-Wound Healing Assay

A total of 1 × 10^5^ HBEC-5i cells were cultured on a 0.1% gelatin-coated 48-well plate in complete medium (DMED/F12, supplemented with 10% heat-inactivated FBS, 1% penicillin and streptomycin, and 30 mg/L ECGS) for 24 h to reach 90% confluence. The endothelial monolayer was scratched using a 200 µL sterile pipette tip. Cells were washed twice gently with PBS and incubated with 120 µL of fresh culture medium without FBS, in the presence or absence of EVs. As a positive control, cells were treated with 50 ng/μL rhVEGF (Peprotech, Rocky Hill, NJ, USA, #100-20B). Each treatment was performed in technical duplicates. Three lines were drawn with a marker on the bottom of each well, perpendicularly to the scratch direction. Images of the scratches at the marked areas were acquired using an inverted optical microscope equipped with a 4× objective, at time 0 h and at 6 h post-scratch. Image analysis was performed using ImageJ 1.53k. The migration index was calculated as the difference between the starting (0 h) and the final (6 h) distance between the migration fronts.

### 4.7. Matrigel Tube Formation Assay

Matrigel Basement Membrane Matrix (Corning, Bedford, MA, USA, #354234) was thawed overnight at 4 °C. Flat-bottom 96-well plates were coated with 80 µL of Matrigel and incubated at 37 °C, 5% CO_2_ for 30 min to allow polymerization. A total of 1.5 × 10^4^ HBEC-5i cells were suspended in 100 µL of assay medium (DMEM:F12 with 2% FBS and 1% penicillin and streptomycin), containing the indicated EVs at the final concentration of 20 µg/mL, and seeded dropwise onto a single well. Each treatment was performed in technical duplicates. For experiments testing the effects of neutralising antibodies against proteins of interest, cells were resuspended in the corresponding treatment media containing EVs that were pre-incubated with anti-ENG (Sigma-Aldrich, Merck Life Science, Milan, Italy, #MABT117), anti-HGF (R&D Systems, Minneapolis, MN, USA, #AF-294-NA), anti-ARTN (R&D Systems, Minneapolis, MN, USA, #MAB2589), or anti-LAP/TGF-β1 (R&D Systems, Minneapolis, MN, USA, #AF-146-NA) antibodies, as indicated, at the final concentration of 5 µg/mL, for 30 min at 37 °C. For all experiments, seeded cells were incubated for 6 h at 37 °C, 5% CO_2_ to allow the formation of cellular networks. Images from at least 3 random areas of each well were acquired with a phase contrast inverted microscope, using a 4× objective. Image analysis was performed using the “Angiogenesis Analyzer” plugin in ImageJ, quantifying parameters related to network complexity.

### 4.8. FITC-Gelatin Degradation Assay

For coverslip preparation, autoclaved 13 mm diameter coverslips were coated with poly-L-lysine (Sigma-Aldrich, Merck Life Science, Milan, Italy, #P4707) and crosslinked with 0.5% glutaraldehyde (Sigma-Aldrich, Merck Life Science, Milan, Italy, #111-30-8), followed by coating with fluorescently labelled gelatin (BD Pharmingen, San Diego, CA, USA, #613186) according to the manufacturer’s instructions. Fluorescence was quenched in 5 mg/mL sodium borohydride (Sigma-Aldrich, Merck Life Science, Milan, Italy, #452882). For the assay, 0.8 × 10^5^ HBEC-5i cells were seeded and cultured o.n. on FITC-gelatin-coated coverslips, following which they were treated in complete medium with 20 μg/mL M-EVs, or with 50 ng/mL rhVEGF. Cells were incubated for 24 h at 37 °C, 5% CO_2_. Cells were fixed in paraformaldehyde (PFA) 4% in PBS and mounted with ProLongTM Gold Antifade mountant (Thermo Fisher Scientific, Eugene, OR, USA#P36934). Images were acquired at 40× magnification through a confocal microscope, Zeiss LSM 800 (Carl Zeiss Microscopy GmbH, Germany). Quantification of digested areas was performed with ImageJ.

### 4.9. BrdU Proliferation Assay

For in vitro proliferation assays, 0.8 × 10^5^ HBEC-5i cells were plated o.n. in a 0.1% gelatin-coated 24-well plate in their standard growth medium. On the assay day, after 2 h of starvation (0% FBS), cells were treated with 10 μM of 5-bromo-2′-deoxyuridine (BrdU, Sigma-Aldrich, Merck Life Science, Milan, Italy, #B5002). Simultaneously, 20 μg/mL of M-EVs were added. Cells were incubated for 6 h at 37 °C, 5% CO_2_. BrdU incorporation was measured by flow cytometry. Briefly, cells were fixed and permeabilised with the BD Cytofix/Cytoperm Fixation/Permeabilization Solution Kit (BD Pharmingen, San Diego, CA, USA, #10482735). Then, cells were treated with 300 μg/mL DNase (Sigma-Aldrich, Merck Life Science, Milan, Italy, #D5025-150KU) for 1 h at 37 °C, followed by incubation with anti-BrdU-AlexaFluor647 (Invitrogen, Carlsbad, CA, USA, #B35140), for 20 min at RT. Labelled cells were detected at FACS Celesta and data analysis was performed with FlowJo software.

### 4.10. Proteome Profiling of EVs

EVs secreted by control and hypoxia-preconditioned microglia were lysed with 4% SDS in PBS 1× and the sample protein concentration was quantified by the Micro BCA Protein Assay Kit (Pierce, Thermo Fisher Scientific, Waltham, MA, USA#23235). Analysis of the angiogenesis-related EV protein cargo was performed using the Proteome Profiler Human Angiogenesis Array Kit (R&D Systems, Minneapolis, MN, USA, #ARY007), according to the manufacturer’s protocol. Briefly, 200 μg protein input was analysed. EV lysates were mixed with 15 µL of biotinylated detection antibody cocktail for 1 h, at RT. Then, the mix was incubated with the nitrocellulose membranes, spotted with specific antibodies against array proteins, o.n. at 4 °C, on a shaker. The following day, membranes were washed from excess antibody–antigen complexes and incubated with the streptavidin–horseradish peroxidase (HRP) for 30 min at RT. For semi-quantitative analysis, membrane images were acquired using the iBright Imaging System (Invitrogen, Carlsbad, CA, USA). Quantification of the mean pixel density of duplicate spots, representing a single angiogenesis-related factor, was performed with ImageJ and expressed as fold induction to the mean value of reference dots present on each membrane, as by the manufacturer’s instructions.

### 4.11. Zebrafish Xenograft

Dechorionized tg(KDRL:GFP) embryos [48] at 48 h post-fertilisation (hpf) were anaesthetised with 0.003% tricaine (Sigma-Aldrich, Merck Life Science, Milan, Italy) and transferred on a Petri dish coated with 3% agarose (Sigma-Aldrich). Isolated EVs were labelled with the Vybrant DiL labelling solution (Invitrogen, Carlsbad, CA, USA) according to the manufacturer’s instructions and resuspended in Matrigel Basement Membrane Matrix (Corning, Bedford, MA, USA, #354234), in an EV/Matrigel volume ratio of 1:3. A volume of 5 nL of each suspension was grafted in the proximity of the sub-intestinal vein (SIV) as previously reported [49], using borosilicate glass capillary needles, and a Pneumatic Picopump connected to a micromanipulator (WPI). At 24 h post-injection (hpi), embryos were fixed in 4% PFA and stained for alkaline phosphatase (AP) activity as described here [49]. The stained embryos were photographed using a Nikon Stereo Microscope SMZ1500 equipped with a DS-Fi1C digital camera. The migration of the SIV branches towards the implant was evaluated by directly counting the number of newly formed sprouting vessels towards the graft area in each injected embryo (*n* = 36 embryos/group for at least two biological replicates).

### 4.12. Mice

CD-1 3-month-old mice (female, *n* = 7 male, *n* = 4) were purchased from Charles River Laboratories (Calco, Italy). Procedures involving animals and their care conformed to institutional guidelines in compliance with national (4D.L.N.116, G.U., suppl. 40, 18 February 1992) and international (EEC Council Directive 2010/63/UE; National Institutes of Health Guide for the Care and Use of Laboratory Animals) law and policies. The protocols were approved by the Italian Ministry of Health (authorizations n°773/2020-PR). Mice were housed in clear plastic cages under a 12 h light/dark cycle and were given ad libitum access to water and food. All efforts were made to minimize the number of animals used and their suffering. In all the experiments, mice were sex- and age-matched, and no further randomization was applied.

### 4.13. Surgery for Stroke Induction and Treatment

The ischemic stroke was induced by performing the distal middle cerebral artery occlusion (dMCAO) technique, as documented by Llovera and colleagues [41]. To obtain a permanent dMCAO animal model, mice were anaesthetized with isoflurane (4% for induction, 1.5–2.5% for maintenance). Combined analgesia, Dexamethasone (2 mg/kg, Dexadreson, MSD Animal Health, Segrate, Italy) and Caprofen (5 mg/kg, Rimadyl, Zoetis, Rome, Italy), was administered through intramuscular and intraperitoneal injection, respectively, before any surgical intervention. During the surgery, body temperature was constantly maintained at ~36 °C using a heating pad (KF Technology, Rome, Italy). The area between ear and eye was shaved and disinfected. Animals were positioned in a lateral position and immobilized with adhesive tape for medical use. Lidocaine cream (2.5%) was applied to the area of interest. A horizontal incision of 4 mm was made between the eye and the ear canal, and from the ear canal a vertical incision of 4–5 mm was made to create a flap of skin that could be retracted. The same incisions were made on the temporalis muscle to expose the skull. A small craniotomy was performed under the decussation of the MCA, the dura was removed with forceps, and the MCA was occluded under the bifurcation with 0.25 mm bipolar forceps (GIMA, Gessate, Italy) connected to an electrosurgical unit (Diatermo 122, GIMA, Gessate, Italy) while the brain tissue remained covered with 0.9% NaCl solution. If spontaneous recanalization occurred, the electrocoagulation was performed again. Finally, the temporalis muscle was relocated and glued in position with veterinary glue (Vetbond, 3M Animal Care Products, St. Paul, MN, USA) and the wound was glued as well. Animals were left recovering in their home cages. After MCAO surgery, mice underwent systemic administration of PBS (for the control group, *n* = 5) or EVs (10 µg in 150 µL PBS, *n* = 6) through intraperitoneal (IP) injections at the injury onset and every 24 h, for a total of 5 consecutive days. On day 7 after stroke induction, mice were perfused and brain tissue collected for subsequent histological evaluations.

### 4.14. Immunofluorescence on Brain Sections

Control and EV-treated animals were deeply anaesthetised with the mix of ketamine–xylazine (100 mg/kg–10 mg/kg) and then transcardially perfused with 20–30 mL PBS, followed by paraformaldehyde (PFA) diluted at 4% in 0.1 m phosphate buffer, pH 7.4. Brains were postfixed for 24 h and then cryoprotected in sucrose (30% in phosphate buffer). Brain coronal sections (60 μm) were obtained using a freezing microtome (Leica, Wetzlar, Germany). Sections were cut in the anteroposterior way, in a serial order, and kept in culture wells, in PBS solution at 4 °C. A total of 4–5 coronal sections were selected to perform immunohistochemical stainings. Free-floating sections were blocked for 1 h at RT with 10% normal goat serum (NGS), 0.3% Triton, and PBS. Slices were then incubated overnight at 4 °C with the indicated primary antibody. Anti-CD31 primary antibody (rat, 1:500, BD Pharmingen, San Diego, CA, USA, #553370) and anti-Ki67 (rabbit, 1:500, Abcam, Cambridge, UK, #15580-100) were diluted in a solution containing 2% NGS, 0.2% Triton, and PBS. The primary antibodies were revealed by incubation for 2 h at RT with a secondary antibody (anti-rat Alexa Fluor 555, anti-rabbit Alexa Fluor 488, 1:500, Invitrogen, Carlsbad, CA, USA), in 1% NGS and 0.1% Triton X-100 and PBS. Images were acquired using a confocal microscope, Zeiss LSM 800, with a 25× objective to assess the microvessel density and proliferating nuclei in the perilesional area (within 500 μm medial and lateral, from the ischemic border). The percentage of the CD31^+^ area was analysed using ImageJ, fixing a threshold for signal positivity for all fields of view, with results expressed as arbitrary units (a.u.), averaged over each section (3 per animal, N = 5 PBS, N = 6 hypEVs) and normalised over PBS controls. Proliferating nuclei were counted for each field of view analysed. The percentage of the Ki67^+^ area was expressed in a.u., fixing a signal threshold in ImageJ, based on 3 sections per animal (N = 3 PBS, N = 4 hypEVs), normalised over PBS controls.

### 4.15. Microglia Isolation

For the MACS-based isolation of microglia, healthy adult mice (C57BL6/J, 3 months old, female *n* = 5, male *n* = 4) were sacrificed, and brains were harvested and kept at 4 °C in HBSS buffer. Microglial cells were freshly isolated using the Neural Tissue Dissociation Kit P (Miltenyi Biotec, Bergisch Gladbach, Germany, #130-092-628). Brains were enzymatically digested at 37 °C, a single cell suspension was made, and myelin debris was removed using the Myelin Removal Beads II Kit (Miltenyi Biotec, Bergisch Gladbach, Germany, #130-096-731) and LS MACS Magnetic Bead Columns (Miltenyi Biotec, Bergisch Gladbach, Germany, #130-042-401) according to the supplier’s instructions. The myelin-free fraction was centrifuged, and cells were resuspended in 90 μL buffer, containing 10 μL CD11b Microglia MicroBeads (Miltenyi Biotec, Bergisch Gladbach, Germany, #130-093-634) at 4 °C for 15 min for positive selection, as per the supplier’s protocol, followed by separation with LS columns. The purity of isolated CD11b^+^ microglial cells (M0) was confirmed by FACS analysis of CD11b and CD45 marker expression. Briefly, 10^5^ cells were collected prior to CD11b MicroBeads separation and post-separation from the CD11b^+^ or CD11b^−^ fractions, and incubated with a mix of anti-CD45-PerCP (BD Pharmingen, San Diego, CA, USA, #557235) and anti-CD11b-APC-AlexaFluor750 (eBioscience, Thermo Fisher Scientific, Wien, Austria, #27-0112-82) antibodies, diluted 1:100 in 100 µL FACS buffer (2% FBS in PBS 1×), for 20 min at 4 °C. Cells were washed with PBS 1× and resuspended in 200 µL FACS buffer, for flow cytometric analysis. Samples were acquired at FACS Celesta and data analysis was performed using FlowJo. Cells isolated from 3 brains were pooled for each in vitro stimulation experiment. Afterwards, 0.5–1 × 10^5^ cells/well were plated in a 24-well plate, pre-coated with 0.01% poly-L-lysine, in DMEM High Glucose Medium (ATCC, Manassas, VA, USA, #30-2002) supplemented with 10% FBS (Gibco, Thermo Fisher Scientific, Paisley, UK, #A5256701), 1% Pen/Strep, and 40 ng/mL mM-CSF (R&D Systems, Minneapolis, MN, USA, #416-ML), refreshing half medium every other day, and were kept in culture for 2–3 weeks, until confluent. Cells were stimulated with the same OGD/R protocol as HMC-3 microglia and were subsequently lysed with 100 µL Trizol per well for RNA analysis.

### 4.16. Statistical Analysis

For in vitro and zebrafish experiments, all data are expressed as mean ± SEM. The Gaussian distribution of data was assessed before performing statistical analysis by applying D’Agostino–Pearson and Shapiro–Wilk normality tests. Differences among groups were evaluated by parametric tests (*t*-test or one-way ANOVA with Tukey’s multiple comparison test) or non-parametric tests (Mann–Whitney test; Kruskal–Wallis test with Dunn’s multiple comparison test), according to sample group numerosity and dataset distribution, as expressed in the figure legends. Statistical significance was set at *p* < 0.05. For in vivo zebrafish experiments, a Chi-Square test was applied. For the proteome profiling analysis of M-EVs, multiple Student’s *t*-test analyses were applied for identifying significant differences in the measured relative pixel density, calculating *p*-values for each protein, across the conditions of interest (hypEVs versus contrEVs). For quicker visualisation, values were represented in a volcano plot, showing the statistical significance (−log(*p*-value)) against the mean pixel density differences calculated in the statistical test. When a *p*-value ≤ 0.05 (cut-off at Y = −log(0.05)) and an absolute mean difference value ≥ 0.05 was found (cut-off at X = 0.05), protein factors were selected for further analysis.

## Figures and Tables

**Figure 1 ijms-25-12508-f001:**
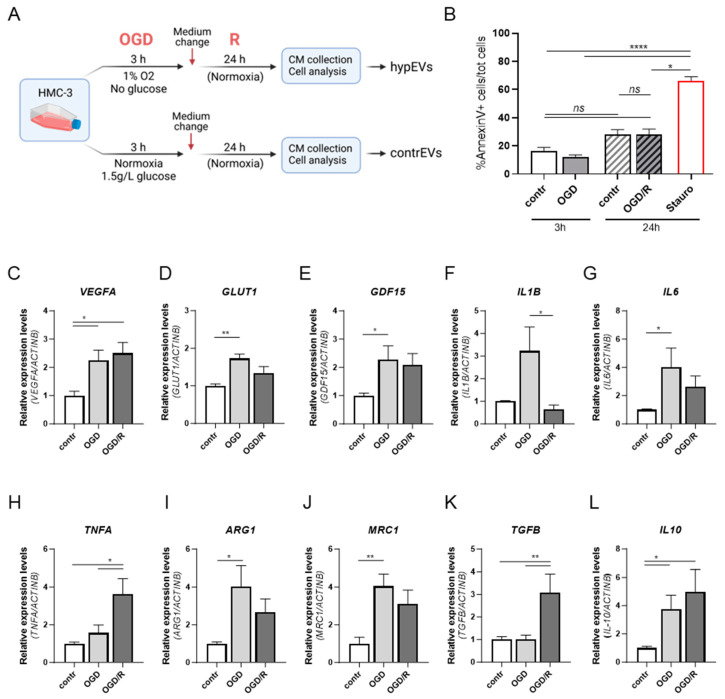
Activation of human microglia in response to OGD/R stimulation. (**A**) Schematic representation of the experimental workflow relative to the stimulation of HMC-3 cells with the OGD/R in vitro protocol. Following exposure to OGD/R, cells were detached for viability and gene expression analyses, while the conditioned medium (CM) was collected to subsequently isolate microglia-derived EVs. (**B**) Flow cytometric analysis of Annexin V in HMC-3 cells. Histograms represent the percentage of apoptotic cells (Annexin V^+^) over the total cells, quantified after OGD (3 h) and OGD/R (24 h) stimulation. Normoxic cells were used as controls at each time point. Cells treated with staurosporine were used as a positive control. Data are reported as relative percentages of apoptotic cells normalised to normoxic controls. (**C**–**L**) qRT-PCR analysis of hypoxia-response genes and microglia activation markers in HMC-3 cells upon OGD/R, confirming microglia activation. Relative gene expression levels were normalised to the housekeeping gene *ACTB*. Bars represent mean ± SEM. N = 3. The Kruskal–Wallis test for multiple comparisons was applied. * *p* < 0.05, ** *p* < 0.01, **** *p* < 0.0001, ns: not significant. CM: conditioned medium; OGD/R: oxygen–glucose deprivation/reperfusion.

**Figure 2 ijms-25-12508-f002:**
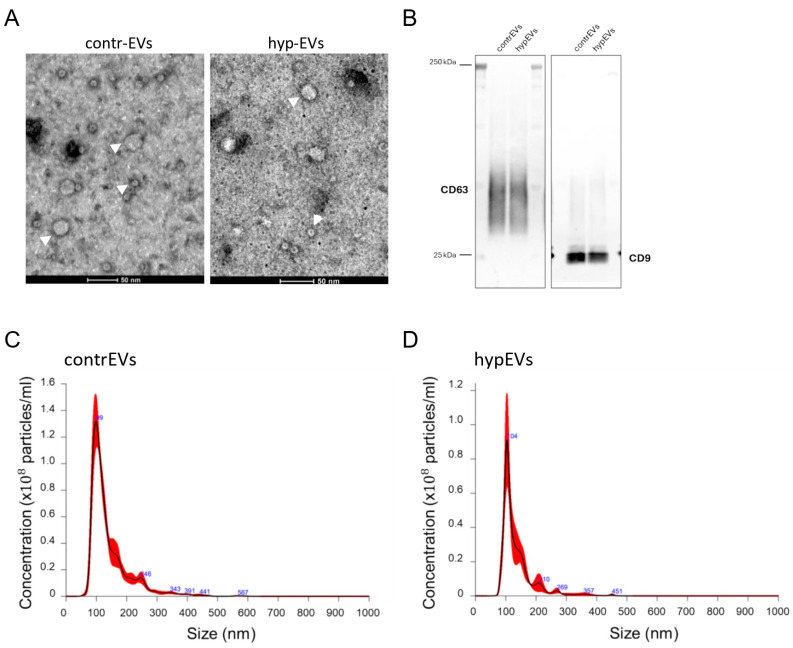
Characterization of EVs isolated from HMC-3-CM. (**A**) Transmission electron microscopy (TEM) images of EVs isolated from contr-HMC-3-CM (**left**) or hyp-HMC-3-CM (**right**). White arrowheads indicate EVs. Scale bar: 50 nm. (**B**,**C**) Western blot of EV lysates, validating the expression of EV-associated proteins CD63 and CD9 in the purified samples. (**C**,**D**) Representative pictures of the nanoparticle tracking analysis (NTA) of EVs isolated by ultrafiltration. The plots represent the size distributions of EVs, displaying the estimated concentration (particles/mL) for each particle size (nm), in both control and hypoxic conditions. The highest peaks and indicated numbers represent the mode vesicle size (99 nm in contrEVs and 104 nm in hypEVs), highlighting an enrichment in vesicles in the small EV (sEV) range (<200 nm).

**Figure 3 ijms-25-12508-f003:**
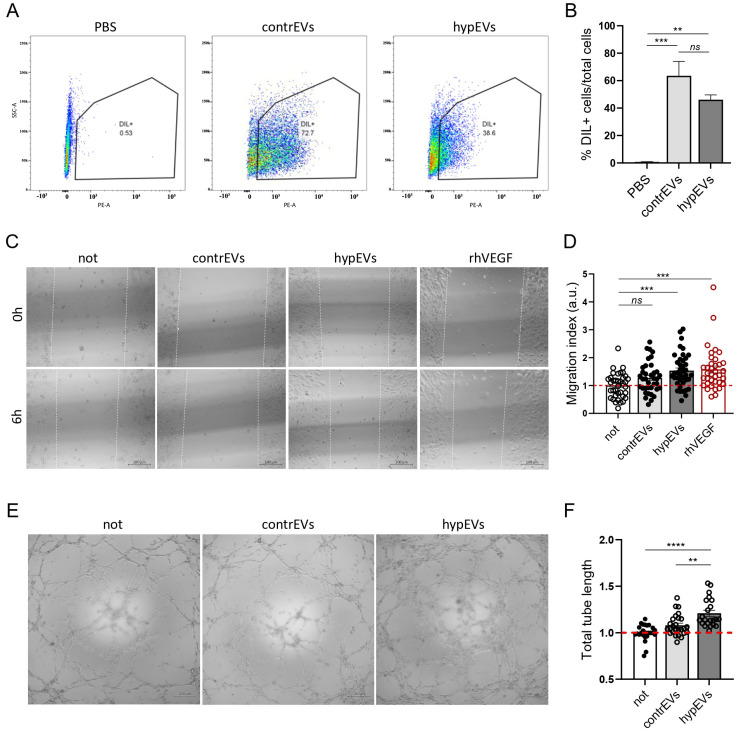
Functional characterisation of microglial EV effects on HBEC-5i brain microvascular cells. (**A**) Uptake of contrEVs and hypEVs by HBEC-5i cells. EVs isolated from contr-HMC-3-CM, hyp-HMC-3-CM, or the PBS vehicle were stained with Dil before treating HBEC-5i cells for 6 h. Representative panels of the gating strategy for cells exhibiting positive fluorescence in the PE (Dil) channel are shown. (**B**) The quantification represents the percentage of Dil^+^ cells after incubation with EVs^Dil^, compared to cells incubated with PBS^Dil^. (**C**,**D**) Scratch wound healing assay performed on HBEC-5i cells. A confluent endothelial cell monolayer was scratched, and cells were cultured in the presence of contrEVs or hypEVs, or with rhVEGF, in serum-free medium. Images of the scratch borders were acquired using an inverted optical microscope with a 4× objective, at time 0 h and after 6 h of culture. Image analysis was performed using ImageJ 1.53k. Results are expressed as a migration index, calculated as the difference between the starting (time 0 h) and the final (time 6 h) distance between the migrating fronts, normalised to the unstimulated control. (**E**,**F**) Tube formation assay performed on HBEC-5i cells. Cells were seeded on Matrigel and cultured for 6 h in the presence of contrEVs, hypEVs, or in medium alone. Images were acquired using a 4× objective. Histograms represent the cumulative tube length of the networks, as quantified using the Angiogenesis Analyzer plugin in ImageJ 1.53k. Scale bar: 100 µm. Results of at least three independent experiments are presented for all assays. Bars represent mean ± SEM. The Kruskal–Wallis test for multiple comparisons with Dunn’s post hoc was applied. ** *p* < 0.01, *** *p* < 0.001, **** *p* < 0.0001, ns: not significant.

**Figure 4 ijms-25-12508-f004:**
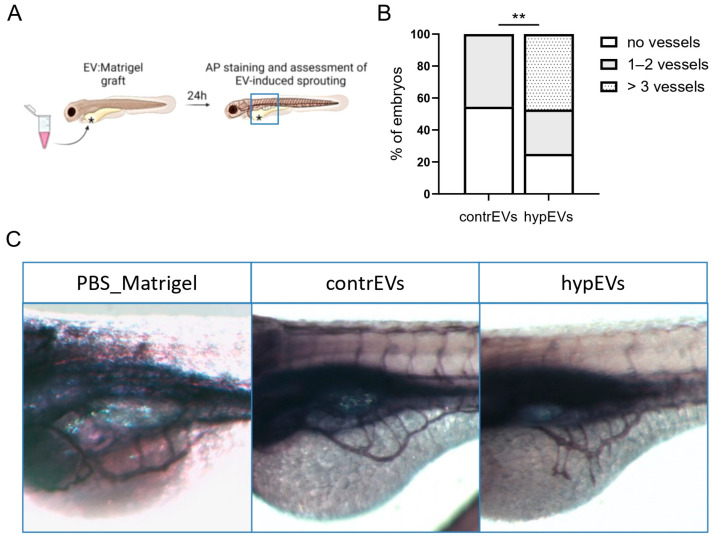
Analysis of the angiogenic potential of microglial EVs in zebrafish. (**A**) Schematic of the experimental set-up. EVs were grafted with Matrigel in the proximity of the developing sub-intestinal vein (SIV), as shown by a black asterisk, and vascular sprouting from the SIV was assessed after 24 h. The area of vessel analysis is represented by a blue square. (**B**) Histograms represent the number of vessels sprouting towards the implant, counted for each embryo, divided into categories (no vessels, 1–2 vessels, >3 vessels). Data are expressed as the percentage of embryos in each category (*n* = 36 embryos/group). (**C**) Representative confocal images showing whole-mount alkaline phosphatase (AP) staining of embryos at 1 dpi in lateral view. Injection of hypEVs stimulated the migration of SIV-derived vascular sprouts towards the implant, while injection with contrEVs resulted in a normal development of the vascular plexus. Chi-Square test was performed. ** *p* < 0.01.

**Figure 5 ijms-25-12508-f005:**
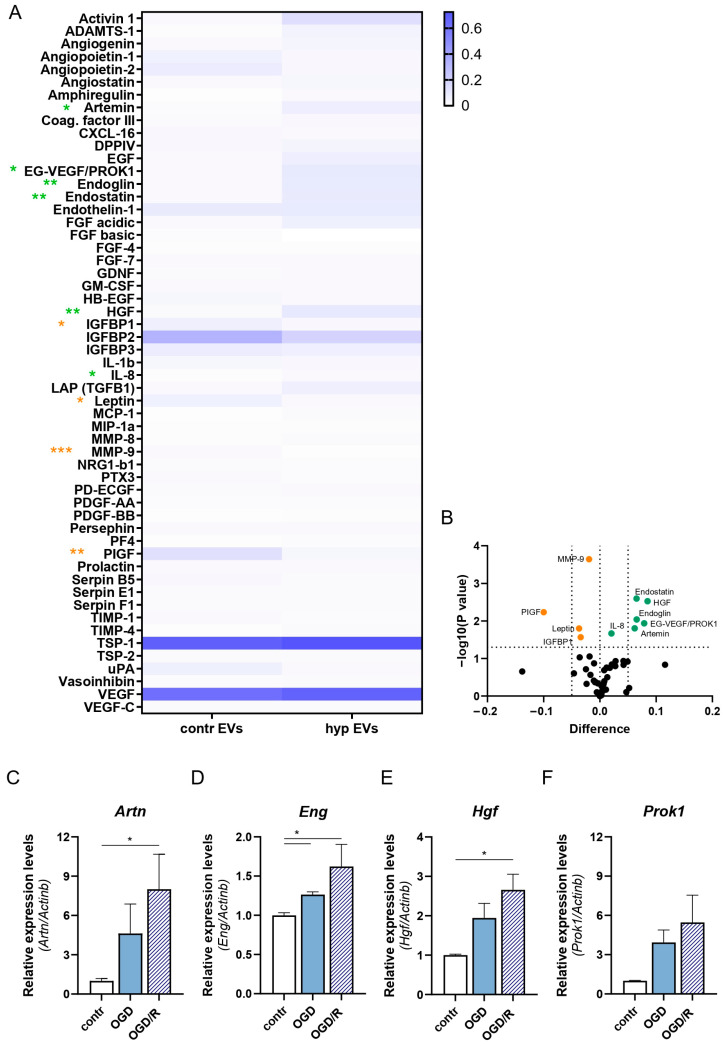
Analysis of the angiogenic protein cargo of hypEVs and contrEVs. (**A**) Proteome profiling analysis was performed on contrEVs and hypEVs. Protein content in EVs was quantified by micro-BCA for equal protein input (200 µg). The mean pixel density for each angiogenesis-related factor, expressed as fold to positive reference dots, is represented in a heat-map. Duplicate analyses of separate EV preparations for each EV type are represented. The non-parametric *t*-test was performed for each factor. Orange asterisks represent factors that are significantly down-regulated (* *p* < 0.05, ** *p* < 0.01, *** *p* < 0.001) in hypEVs. Green asterisks represent factors that are significantly enriched (* *p* < 0.05, ** *p* < 0.01) in hypEVs. (**B**) The volcano plot reports the −log(*p*-value) (Y axis), graphed against the mean pixel density differences (X axis). Orange dots represent factors that are significantly down-regulated (*p* < 0.05) in hypEVs. Green dots represent factors that are significantly enriched (*p* < 0.05) in hypEVs. (**C**–**F**) Primary mouse microglial cells were isolated from the brain of adult healthy mice and stimulated in vitro with the OGD/R protocol. The mRNA levels of the identified pro-angiogenic markers (*Artn*, *Eng*, *Hgf*, and *Prok1*) were quantified by qRT-PCR. Expression levels are normalised to the house-keeping gene *Actb.* N = 3. Bars represent mean ± SEM. The Kruskal–Wallis test was applied (**C**–**F**), * *p* < 0.05.

**Figure 6 ijms-25-12508-f006:**
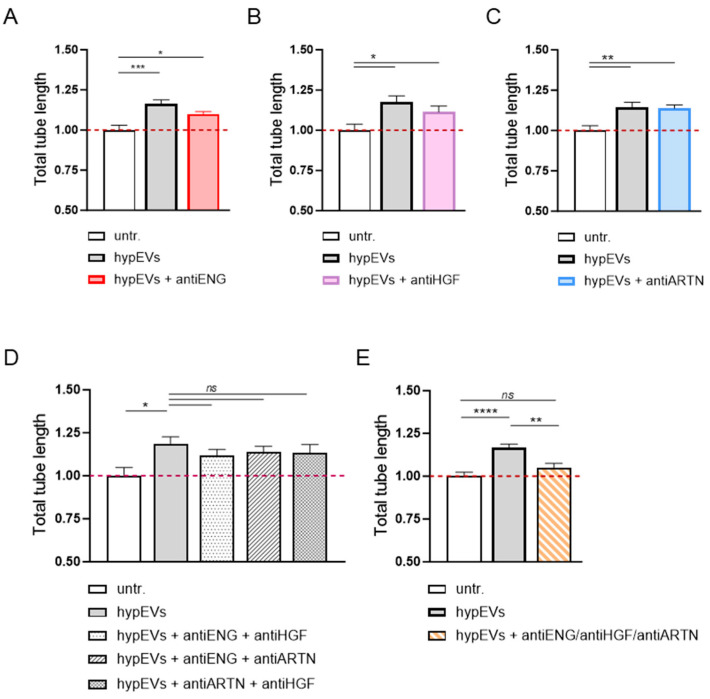
Role of ENG, HGF, and ARTN in mediating hypEVs’ pro-angiogenic effect towards ECs. (**A**–**E**) Results of tube formation assays performed on HBEC-5i cells. Cells were seeded on Matrigel and cultured in a medium containing hypEVs (20 µg/mL), pre-incubated with blocking antibodies. Specifically, anti-ENG, anti-HGF, anti-ARTN antibodies, or antibody combinations were used, as indicated. Capillary-like networks were imaged after 6 h, using a 4× objective. Histograms represent the cumulative tube length of the networks, as quantified using the Angiogenesis Analyzer plugin in ImageJ. N = 3. Bars represent mean ± SEM. The Kruskal–Wallis test was applied. * *p* < 0.05, ** *p* < 0.01, *** *p* < 0.001, **** *p* < 0.0001, ns: not significant.

**Figure 7 ijms-25-12508-f007:**
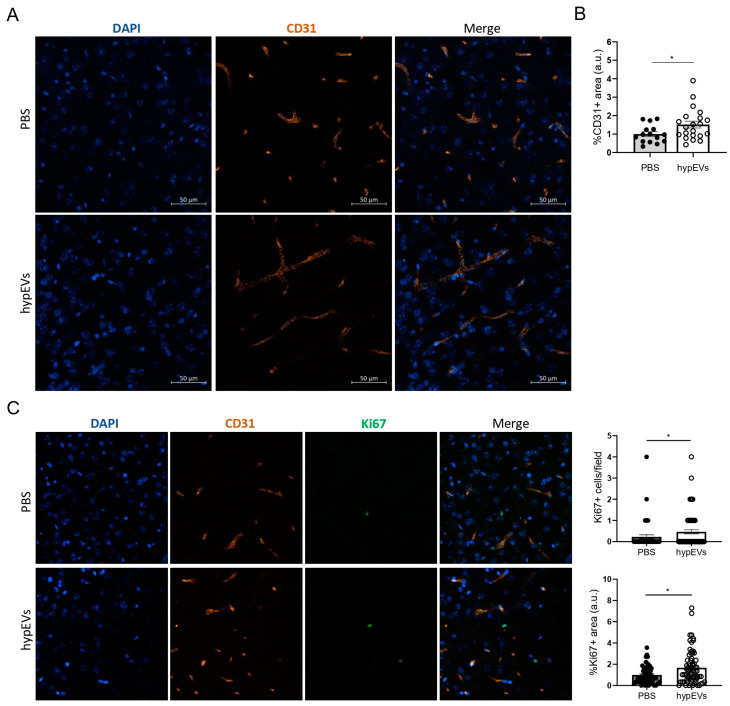
Evaluation of the hypEV effects on brain angiogenesis in a mouse model for ischemic stroke. After MCAO, mice were injected intraperitoneally with PBS or hypEVs, immediately after the procedure and each 24 h, for a total of 5 consecutive days. Mice were perfused at day 7 post-MCAO (*n* = 5 PBS, *n* = 6 hypEVs). (**A**) Representative pictures of blood vessels in the perilesional area. Immunofluorescence on brain sections was performed using anti-CD31 antibodies (endothelial cells) and Hoechst (nuclei). Images were acquired at a confocal microscope, using a 25× objective. Scale bar: 50 µm. (**B**) Quantification of the CD31 signal in the perilesional region was performed using ImageJ, fixing a threshold to detect the percentage of the CD31^+^ area, expressed as arbitrary units (a.u.), for each field of view analysed. (**C**) Analysis of proliferating nuclei in the perilesional area. The number of proliferating cells per field of view analysed is reported. Quantification of the Ki67^+^ area was performed using ImageJ and is expressed as normalised to controls. Bars represent mean ± SEM. The non-parametric *t*-test was applied. * *p* < 0.05.

**Table 1 ijms-25-12508-t001:** Primers used for qRT-PCR analysis. FW (Forward); RV (Reverse); h (human); m (mouse).

Gene	Target Species	Primer Nucleotide Sequence 5′ > 3′	
*GDF15*	h	FW	CCTGAGACACCCGATTCCTG
	h	RV	CCCAAGAAGGTCACCCCAAT
*VEGFA*	h	FW	CTGGAGCGTGTACGTTGGT
	h	RV	CGTTTAACTCAAGCTGCCTCG
*GLUT1*	h	FW	ATGGGCTTCTCGAAACTGGG
	h	RV	CCGCAGTACACACCGATGAT
*IL1B*	h	FW	CTGTCCTGCGTGTTGAAAGA
	h	RV	TTGGGTAATTTTTGGGATCTACA
*IL6*	h	FW	AACCTGAACCTTCCAAAGATGG
	h	RV	TCTGGCTTGTTCCTCACTACT
*TNFA*	h	FW	ATGAGCACTGAAAGCATGATCC
	h	RV	GAGGGCTGATTAGAGAGAGGTC
*ARG1*	h	FW	CGGACTGGACCCATCTTTCA
	h	RV	CCGAGCAAGTCCGAAACAAG
*MRC1*	h	FW	ACCTGCGACAGTAAACGAGG
	h	RV	TGTCTCCGCTTCATGCCATT
*TGFB1*	h	FW	GGACTGCGGATCTCTGTGTC
	h	RV	CTGGGCTTGTTTCCTCACCT
*IL10*	h	FW	CACATCAGGGGCTTGCTCTT
	h	RV	GGTGCAGCTGTTCTCAGACT
*ENG*	h	FW	CCATCCTTCGGACAGCAACT
	h	RV	CAGGCTGGAATTGTAGGCCA
*HGF*	h	FW	GCAATTAAAACATGCGCTGACA
	h	RV	TCCCAACGCTGACATGGAAT
*ARTN*	h	FW	CTGCCAAGGCCACACTTTTG
	h	RV	CTTCTAGGCACCTTTCCGGG
*PROK1*	h	FW	ATGCTCCTCCTAGTAACTGTGTCTGACT
	h	RV	CTCGAAGCCACAGGCTGATG
*ACTB*	h	FW	CATGTACGTTGCTATCCAGGC
	h	RV	CTCCTTAATGTCACGCACGAT
*Eng*	m	FW	GCTGGAGTCGTAGGCCAAGT
	m	RV	CTGCCAATGCTGTGCGTGAA
*Hgf*	m	FW	ATCCCAAATCGTCCTGGTATTT
	m	RV	CTGGCCTCTTCTATGGCTGGCTATTAC
*Artn*	m	FW	GGACCCCTTTGGTATGGAGTG
	m	RV	GTGGGACAATGCAGTAGGCT
*Prok1*	m	FW	GTGAACGAGATATCCAGTGTGGG
	m	RV	GTTGGCGTTTCCTCAAGAAGGG
*Actb*	m	FW	AGTGTGACGTTGACATCCGTA
	m	RV	GCCAGAGCAGTAATCTCCTTC
*Tgfb1*	m	FW	CACAGTACAGCAAGGTCCTTGC
	m	RV	AGTAGACGATGGGCAGTGGCT

## Data Availability

The original contributions presented in the study are included in the article/Appendix A; further inquiries can be directed to the corresponding author.

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
