# Peer review of "Hypoxic Human Microglia Promote Angiogenesis Through Extracellular Vesicle Release"

_ijms, 2024, doi:10.3390/ijms252312508_

Round 1

Reviewer 1 Report (Previous Reviewer 2)

Comments and Suggestions for Authors

The authors made changes improving the quality of the paper

Reviewer 2 Report (Previous Reviewer 1)

Comments and Suggestions for Authors

I have no comments to the presented version of manuscript.

This manuscript is a resubmission of an earlier submission. The following is a list of the peer review reports and author responses from that submission.

Round 1

Reviewer 1 Report

Comments and Suggestions for Authors

In this study, Testa et al. aim to clarify the molecular mechanisms underlying microglia-endothelial cell interactions under ischemic stroke conditions. Using both in vitro and in vivo models, the authors investigated the angiogenic signatures and properties of extracellular vesicles (EVs) released by human microglia following hypoxia-reperfusion stimulation.

Understanding this mechanism could provide valuable insights into the role of microglia in ischemic stroke, potentially improving diagnosis, prognosis, and monitoring in ischemic diseases. While the authors have gathered an impressive amount of data and performed elegant in vitro work, I have a few questions regarding the in vivo aspects of the presented experiments:

1.      Were normoxic EVs transplanted into the ischemic brain? If so, what effect did they have?

2.      What was the rationale for using peritoneal injection as the method for EV transplantation? Given that intravenous injections often result in a large number of EVs being distributed to other organs such as the liver, lungs, or intestines, have you examined the distribution of intraperitoneally injected EVs in these organs? Do you believe intraperitoneal injections are more effective than intravenous ones in this context?

3.      Have you attempted labeling the EVs prior to transplantation in the ischemic model to track their distribution or activity?

4.      The in vivo portion of the manuscript seems relatively modest. Have you considered assessing the angiogenic potential of the transplanted EVs using additional markers or approaches? Additionally, did these injections have any measurable effect on the infarct area in the ischemic brain?

Reviewer 2 Report

Comments and Suggestions for Authors

Testa and colleagues report the enhancement of angiogenesis in human brain endothelial cells culture treated with extravesicular vesicles (EV) derived from hypoxic human microglial cells. After hypoxia/reoxygenation, the microglial cells exhibited the enhanced expression of several anti- and proinflammatory genes, as well as VEGFA. The proteomic analysis has shown the increased presence of endoglin, artemin and HGF in hypoxic microglial EVs, and simultaneous treatment of EVs with blocking antibodies to these three proteins significantly decreased the pro-angiogenic effect of EVs. Besides in vitro experiments using a brain endothelial cell culture, the authors also demonstrated the angiogenic effects of human glial cells-derived EVs in zebrafish and in a mouse model of ischemic stroke. This is an interesting study, the first one to show the angiogenic effect of microglia-derived EVs using both human glial cells and human brain endothelial cells.

Critiques:

1.      Control EVs are missing in Figure 6.

2.      The expression of angiogenesis-related genes in endothelial cells stimulated with EVs has not been studied.